# A Large Cluster of New Onset Autoimmune Myositis in the Yorkshire Region Following SARS-CoV-2 Vaccination

**DOI:** 10.3390/vaccines10081184

**Published:** 2022-07-26

**Authors:** Gabriele De Marco, Sami Giryes, Katie Williams, Nicola Alcorn, Maria Slade, John Fitton, Sharmin Nizam, Gayle Smithson, Khizer Iqbal, Gui Tran, Katrina Pekarska, Mansoor Ul Haq Keen, Mohammad Solaiman, Edward Middleton, Samuel Wood, Rihards Buss, Kirsty Devine, Helena Marzo-Ortega, Mike Green, Dennis Gerald McGonagle

**Affiliations:** 1NIHR Leeds Biomedical Research Centre, Leeds Teaching Hospitals NHS Trust, Chapel Allerton Hospital, Leeds LS7 4SA, UK; g.demarco@leeds.ac.uk (G.D.M.); kirsty.devine@nhs.net (K.D.); h.marzo-ortega@leeds.ac.uk (H.M.-O.); 2Section of Experimental Rheumatology, The Leeds Institute for Rheumatic and Musculoskeletal Medicine, University of Leeds, Leeds LS7 4SA, UK; S.Giryes@leeds.ac.uk; 3York and Scarborough Teaching Hospitals NHS Foundation Trust, York YO31 8HE, UK; katie.williams@york.nhs.uk (K.W.); nicola.alcorn@york.nhs.uk (N.A.); maria.slade@nhs.net (M.S.); john.fitton@york.nhs.uk (J.F.); 4Mid Yorkshire Hospitals NHS Trust, Wakefield WF1 4DG, UK; sharmin.nizam1@nhs.net (S.N.); gayle.smithson@nhs.net (G.S.); khizer.iqbal1@nhs.net (K.I.); 5Harrogate and District NHS Foundation Trust, Harrogate HG2 7SX, UK; gui.tran@nhs.net (G.T.); katrina.pekarska@nhs.net (K.P.); mike.green4@nhs.net (M.G.); 6Bradford Teaching Hospitals NHS Foundation Trust, Bradford BD9 6RJ, UK; mansoorulhaq.keen@nhs.net; 7Hull University Teaching Hospitals NHS Trust, Hull HU3 2JZ, UK; mohammad.solaiman1@nhs.net (M.S.); edward.middleton3@nhs.net (E.M.); samuel.wood8@nhs.net (S.W.); rihards.buss@nhs.net (R.B.)

**Keywords:** myositis, COVID vaccination, adjuvanticity

## Abstract

Background: The novel SARS-CoV-2 vaccines partially exploit intrinsic DNA or RNA adjuvanticity, with dysregulation in the metabolism of both these nucleic acids independently linked to triggering experimental autoimmune diseases, including lupus and myositis. Methods: Herein, we present 15 new onset autoimmune myositis temporally associated with SARS-CoV-2 RNA or DNA-based vaccines that occurred between February 2021 and April 2022. Musculoskeletal, pulmonary, cutaneous and cardiac manifestations, laboratory and imaging data were collected. Results: In total, 15 cases of new onset myositis (11 polymyositis/necrotizing/overlap myositis; 4 dermatomyositis) were identified in the Yorkshire region of approximately 5.6 million people, between February 2021 and April 2022 (10 females/5 men; mean age was 66.1 years; range 37–83). New onset disease occurred after first vaccination (5 cases), second vaccination (7 cases) or after the third dose (3 cases), which was often a different vaccine. Of the cases, 6 had systemic complications including skin (3 cases), lung (3 cases), heart (2 cases) and 10/15 had myositis associated autoantibodies. All but 1 case had good therapy responses. Adverse event following immunization (AEFI) could not be explained based on the underlying disease/co-morbidities. Conclusion: Compared with our usual regional Rheumatology clinical experience, a surprisingly large number of new onset myositis cases presented during the period of observation. Given that antigen release inevitably follows muscle injury and given the role of nucleic acid adjuvanticity in autoimmunity and muscle disease, further longitudinal studies are required to explore potential links between novel coronavirus vaccines and myositis in comparison with more traditional vaccine methods.

## 1. Introduction

The “Severe Acute Respiratory Syndrome-related Coronavirus type 2” (SARS-CoV-2) infection has resulted in over 5 million deaths and numerous other medical and societal issues. In addition to natural resistance and infection-acquired immunity, vaccines represented a fundamental element in mitigating against severe “Coronavirus Disease 2019” (COVID-19) [1,2,3]. The authorised COVID-19 vaccines have shown efficacy, safety, and tolerability in both randomised clinical trials (RCTs) and in the real-world setting [1,2,3,4,5,6]. Many of these vaccines are based on novel strategies based around DNA and RNA technology and some rare—though potentially serious—autoimmune diseases have emerged within days or weeks of vaccine utilisation, including vaccine-induced immune thrombotic thrombocytopenia (VITT) with DNA vaccines [7] and myopericarditis with RNA vaccines [8].

Many immune-mediated diseases (IMDs) are characterised by the emergence of autoantibodies several months or even years before the clinical onset and presentation [9]. The adjuvanticity of the available SARS-CoV-2 vaccines is novel and at least in part depends on the intrinsic vaccine messenger RNA (mRNA) or DNA. Both stimulate innate immunity through endosolic and cytoplasmic nucleic acid receptors—such as Toll-Like Receptors (TLRs) 3, 7, 8, and 9—as well as components of the inflammasome, including Retinoic acid-Inducible Gene I (RIG-I) and Melanoma Differentiation-Associated Gene 5 (MDA5) [10,11].

A role for altered nucleic acid metabolism with aberrant TLR pathway activation has been postulated in both experimental and human autoimmune connective tissue diseases [12,13]. Given that intramuscular vaccination is capable of releasing muscle antigens and the RNA/DNA vaccine components may get taken up in the muscle [14], then a theoretical basis for emergent muscle autoimmunity exists which is distinct from conventional vaccines that are thought to act mainly on antigen presenting cells and other immune cells. Bearing these theoretical considerations in mind, we noted emergent autoimmunity in our region and especially an unexpected high number of new onset myositis cases temporally linked to SARS-CoV-2 vaccination in the Yorkshire region of the United Kingdom (UK).

## 2. Methods

This study was reported according to the “CAse REports” (CARE) guidelines [15]. All participants recruited granted verbal or written consent to the local treating physicians for the use of their anonymised data. Six NHS Trusts from Yorkshire and Humber region in the UK, (Bradford Teaching Hospitals NHS Foundation trust; Harrogate and District NHS Foundation Trust; Hull University Teaching Hospitals NHS Trust, Leeds Teaching Hospitals NHS Trust; Mid Yorkshire Hospitals NHS Trust and York and Scarborough Teaching Hospitals NHS Foundation Trust—see map in Figure 1) provided data. The authors evaluated patients presenting and diagnosed with: (a) myositis; (b) occurring in plausible temporal relationship with any agent authorised in UK for the SARS-CoV2 vaccination programme.

Diagnosis of myositis was based on referring consultant overall clinical opinion, informed by presentation, symptoms, laboratory findings (creatine-kinases (CK); C-reactive protein, CRP; auto-antibodies), magnetic resonance imaging (MRI) and biopsy—where available. No specific time frame after vaccination agent exposure was specified, as it was felt as too narrow to restrict myositis onset to one month or less since it is well established that autoantibodies precede the clinical onset of IMIDs by several months or years [16].

As suggested by World Health Organization (WHO) guidance [17], Adverse Events Following Immunization (AEFI) represent untoward medical events following vaccination. According to WHO, vaccines could represent the primary factor linked to AEFI or cofactors within complex events [18], although this approach has shortcomings, but robust alternatives are lacking [19]. After validating myositis diagnosis and excluding non-vaccination-related causes, biological plausibility and temporal compatibility between the immunization and the occurrence of the AEFIs were assessed as previously described [20]. Descriptive statistics were used to summarize variables including age, gender, history of IMIDs, average time to onset of symptoms, severity of disease course, therapeutics administered, and key clinical and laboratory findings.

## 3. Results

The estimated Yorkshire population size is 5.6 million and is ethnically diverse. Of 19 cases referred for evaluation, 15 new cases fulfilled the selection criteria (new presentation of myositis—as diagnosed by the treating clinician—occurring in plausible temporal relationship with any COVID-19 vaccine). The remaining 4 cases were excluded for the following reasons: post-vaccine flare of already established myositis (3 cases); myositis onset potentially preceding vaccine exposure.

Overall, 10 cases were females, 5 were males. The average age at myositis symptoms onset was 66.1 years (standard deviation 13.7; range 37–83). Vaccination agents recorded before the onset of myositis were AZD1222/ChAdOx1 (9 cases) and BNT162b2 (6 cases). The median time between exposure to vaccination agent and myositis onset was 5 weeks (interquartile range 4–9; absolute range 1–34 weeks). Myositis occurred in 5 cases after dose 1; in 7 cases after dose 2; in 3 cases after dose 3. In total, 6 out of 15 cases were on statins at the time of myositis onset.

Myositis symptoms are described in Table 1. In one case, CKs were normal and biopsy normal (amyopathic variant). No cases had evidence of a paraneoplastic myositis. The most common non-muscular clinical manifestation was skin involvement (only in 3 cases) and interstitial lung disease (3 cases, representative case in Figure 2). All 4 cases of DM had received DNA vaccination.

Anti-nuclear autoantibodies (ANA) were positive in 8 cases and negative in 7 cases. Myositis-associated autoantibodies were positive in 10 cases and negative in 5 cases. Case 9 was reported as positive for anti-Ro/RNP and anti-chromatin autoantibodies prior to exposure to vaccines and myositis development. This patient had previously been treated as an inflammatory arthritis and was treated with azathioprine years before vaccination and could represent an undefined connective tissue disease prior to definitive myositis diagnosis post vaccination.

Muscle biopsy (case 10 sample is shown in Figure 3) was performed in 10 cases: 5 were compatible with polymyositis, 3 with necrotising features, 1 compatible with dermatomyositis and 1 histologically suggestive of mitochondrial myopathy, but the clinical picture was of autoimmune myositis. Overall, 4 out of 15 patients had classical DM rash pointing towards a predominant DM picture. Of note, in 3 cases (20%), investigations revealed anti 3-hydroxy-3-methylglutaryl-coenzyme A reductase antibodies (all on statins). Magnetic resonance imaging (MRI) of thigh muscle masses (Figure 3) was performed in 11 cases and intramuscular oedema was the most prominent imaging feature (11/11 MRI).

In total, 13 cases were treated with corticosteroids and/or immunosuppressants (12 cases) as detailed in Table 1. With the exception of case 12, patients described in this series improved (both in amelioration of strength and CK level reductions after treatment, although time-to-recovery and degree of recovery differed across cases (Table 2). Complications such as aspiration pneumonia, cataract, pulmonary embolism, anuria, respiratory arrest and stroke occurred in 5 cases (Table 1).

## 4. Discussion

The incidence of autoimmune myositis ranges in 1.16 to 19/million/year according to estimates [21]. Herein, we report a cluster of autoimmune myositis that was temporally related to SARS-CoV-2 vaccination in the Yorkshire region of UK with 40% of cases on statins and with three of these associated with Anti-HMGCR autoantibodies. In this series, myositis followed immunization with both DNA-based and RNA-based vaccines. Time-to-symptoms onset was variable across the cases reported, as well as the association to vaccine cycles timings (that is, occurrence after dose one; dose two or occurrence after dose three—implemented in UK since November 2021). Some of the cases reported could be classified as polymyositis—historically the most common of idiopathic inflammatory myositidies [22].

Our voluntary reporting strategy has limitations, including selection bias (namely, reporting of more severe and early-stage cases). It is also important to acknowledge that the temporal links between vaccination doses administered and the development of myositis-related symptoms was unclear. In Yorkshire, there was relatively little or no exposure to mRNA-1273 (Moderna) or JNJ-78436735 (Ad26.COV2.S) SARS-CoV-2 vaccines. Furthermore, it is unclear what the optimal cut off time is for establishing putative links between vaccines and potential autoimmunity. In the context of musculoskeletal disorders, reactive arthritis typically occurs within weeks to a month of infection or vaccine challenge, but it is important to recognise that muscle autoantibodies can precede autoimmune disease by several years [15]. Nevertheless, most of our cases occurred in a short timeframe following vaccination, but active surveillance in the coming years for evidence of autoimmune myositis development at a later timeframe may be warranted.

Our findings could represent mere coincidence, but collectively, the number of new myositis cases was in considerable excess of our normal clinical expectations. Given that autoimmune myositis is associated with autoantigens that mostly often bind to native RNA, whereas autoimmunity in systemic lupus erythematosus (SLE) is linked to autoantigens bound to DNA, we were interested to see whether a particular association between DNA or RNA vaccination and myositis was evident. We found numerically more cases of myositis in cases exposed to the AZD1222/ChAdOx1 vaccine; however, this finding could be due to the prominent role that the AZD1222/ChAdOx1 agent had in the initial vaccine rollout in UK (in other countries—such as Germany and Israel—the BNT162b2 agent was more commonly administered). Our data do not show specific tendency of vaccine cycle positioning, as myositis cases occurred almost equally after dose 1 or 2. No signal was detected in regards with potential role of dose 3 (3 cases in total), which argues against the idea of autoimmunity that might be expected to be worse with repeated booster injections following earlier breakage of immune tolerance.

One small series of 3 auto-immune myositis occurring after immunization with agent AZD1222/ChAdOx1 is available in the medical literature [23]. In addition, there has also been sporadic case reports of myositis following DNA and RNA vaccines [24,25]. Patients described in our series underwent vaccination cycles with agents approved in UK at the time of clinical presentation and data collection (that is, BNT162b2; AZD1222/ChAdOx1). Given that the intra-muscular injection technique may cause muscle injury, the consequent release of naturally segregated intra-cellular, muscle-specific (auto) antigens could ultimately lead to abundant presentation activity in regional lymph node draining sites of vaccine inoculation (Figure 4A). Combined with potential uptake of RNA/DNA into myocytes and other cell types, antigen presentation by myocytes might favor autoimmunity over conventional adjuvants with the emergence of T-cells and their return to the muscle compartment (Figure 4B) [26]. This mechanism would offer a plausible explanation for muscle autoimmunity, especially polymyositis, which is more muscle-centric compared with dermatomyositis which is more muscle-interstitium- and vascular-centric. We noted numerically more polymyositis cases, but these findings need confirmation in larger cohorts and with robust longitudinal epidemiological surveys. Both RNA and DNA vaccines are also associated with excellent potentiation of both CD4+ and CD8+ T-cell reactivity, with both cell types being incriminated in autoimmune myositis [27] (Figure 4B).

To conclude, it is our impression that a surprisingly high rate of new onset myositis following novel RNA and DNA SARS-CoV-2 vaccination recently emerged in the Yorkshire region. We have reported a temporal association between SARS-CoV-2 vaccines and myositis onset and in the light of a mass vaccine campaign, this does not imply causality. Proof of a causal association is lacking. However, whilst this could be mere co-incidence, the link between muscle injury and novel adjuvants priming of CD4+ and CD8+ T-cells that is known to occur after vaccination merits consideration. A specific epidemiological evaluation for autoimmune muscle disease is needed in the post SARS-CoV-2 era and mechanistic studies of novel vaccines and muscle autoimmunity may address this issue further.

## Figures and Tables

**Figure 1 vaccines-10-01184-f001:**
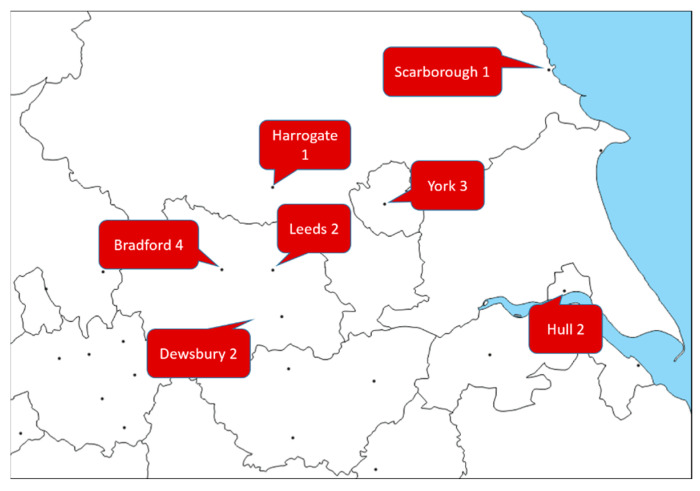
Map highlighting the distribution of new-onset myositis cases across the Yorkshire region, adapted from https://d-maps.com/carte.php?num_car=18610&lang=en (accessed on 12 May 2022). Dewsbury and District Hospital is part of The Mid Yorkshire Hospitals NHS Trust, alongside Pinderfiels Hospital of Wakefield.

**Figure 2 vaccines-10-01184-f002:**
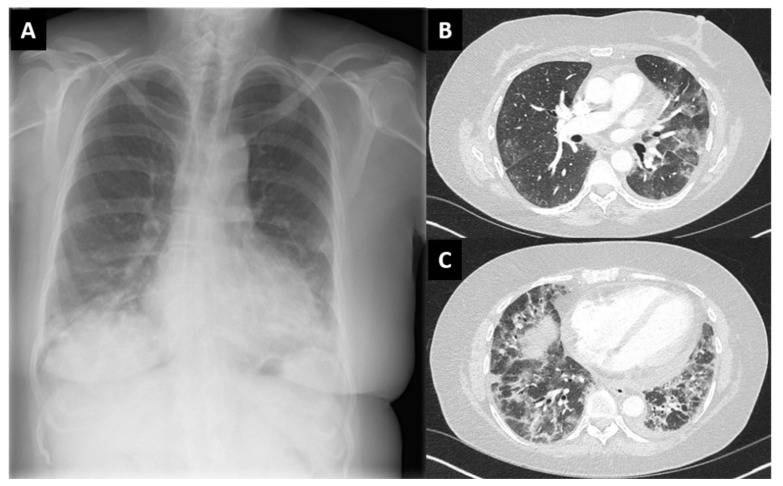
Chest X-ray (**A**) and high-resolution computed tomography scan (**B**,**C**) of case 3 (female, 58 years old) with findings compatible with interstitial lung disease. Lung biopsy not performed.

**Figure 3 vaccines-10-01184-f003:**
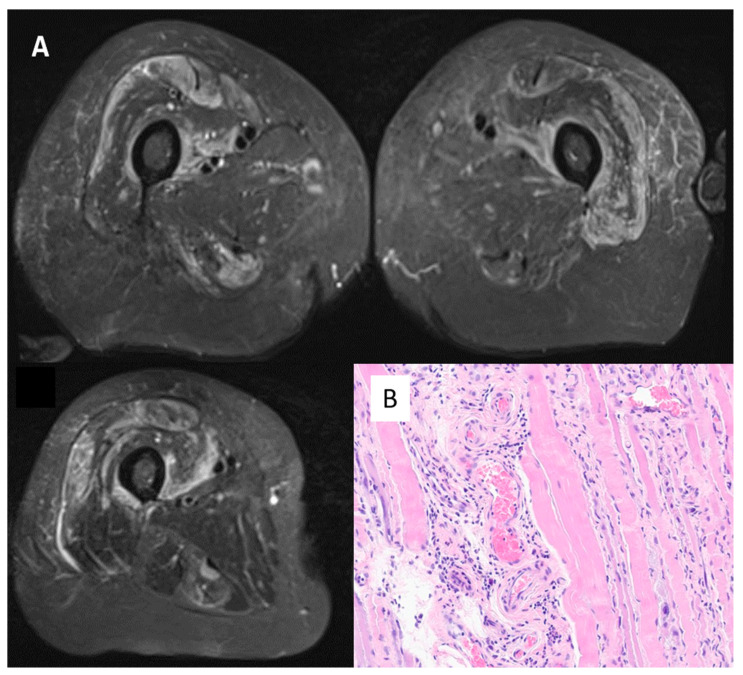
Panel (**A**)—MRI scan of thighs from Case 1 (female, 68 years old). Oedema in the quadriceps (vastus intermedium femuri; vasti medialis and rectum femuri) bilaterally, pointing to inflammation in the muscle masses explored. Panel (**B**)—Muscle biopsy from case 10 (female, 71 years old). Haematoxylin/Eosin, magnification 400×. Perimyseal pathology with local thrombosis consistent with a vasculopathy and macrophages present consistent with immune-mediated necrotising myositis. Anti 3-hydroxy-3-methylglutaryl-coenzyme A reductase antibodies were positive.

**Figure 4 vaccines-10-01184-f004:**
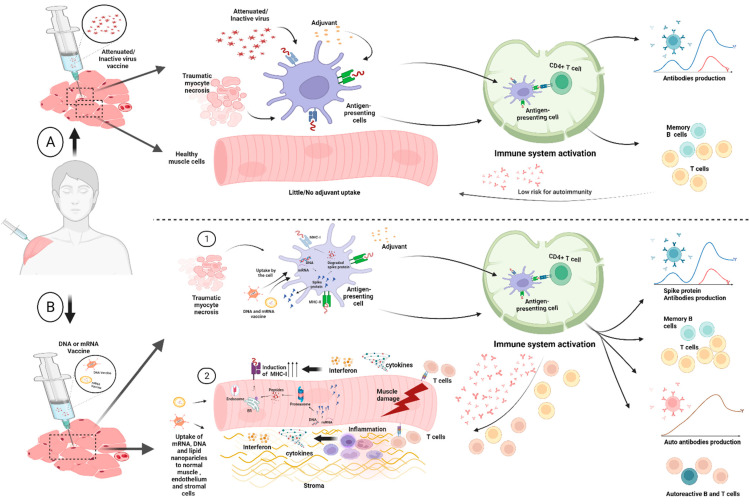
Panel (**A**): Conventional vaccines involve the delivery of a killed pathogen, attenuated pathogen or a protein subunit with adjuvants—including alum and others—to the inoculation site. This is associated with local muscle and endothelial and supporting tissue injury at the site of vaccination. It is thought that uptake of both the antigens and adjuvants by the antigen presenting cells leads to activation of such cells and migration to the regional lymph nodes where lymphocyte priming takes place and robust antibody responses to antigen takes place. In theory, vaccine-associated injury from the injecting needle might release self-antigens that are also taken up by the Antigen-Presenting Cells (APCs) and could theoretically lead to tolerance failure and autoimmunity. However, this is not something that is recognised in the clinical setting with conventional vaccines; hence, it seems unlikely to occur. Panel (**B**): Conventional vaccines the novel DNA- and RNA-based vaccines are also taken up by antigen-presenting cells and migrate to regional lymph nodes and likewise damaged self-muscle, endothelial and stromal tissue elements from sites of injury could undergo pinocytosis and likewise be transported. Copying elements of the viral life cycle within APCs facilitates CD4 and cytotoxic CD8 T-cell responses, that are superior to conventional vaccines and could thus contribute also to the development of better cell-mediated immune responses [28,29]. There is actually a very limited amount of data about the uptake and processing of nucleic acid vaccines following intramuscular injection; however, direct uptake of RNA and DNA into the muscle has been reported [26,30,31]. Further experimental studies and epidemiological surveys are needed to test this hypothesis. Image created with BioRender.com.

**Table 1 vaccines-10-01184-t001:** Clinical, laboratory and Imaging Features of Myositis Cases.

Centre Gender Age (in Years, at Myositis Onset)	Past Medical History for Autoimmunity	SARS-CoV-2 Vaccine. (Time between Exposure and Myositis Onset, in Weeks)	Details Related to Myositis Including Muscle Related Autoantigen (MRA)	Non-Muscular Manifestations (Complications, if Occurred)	Treatment
Case 1	No	AZD1222/ChAdOx1	Asthenia, dysphonia, impaired swallowing. CK 536 U/L and CRP 55 mg/L, no biopsy. Widespread quadriceps oedema on MRI ANA +, Myoblot positive (SAE-1)	Gottron’s papules; heliotrope rash; shawl’s sign (aspiration pneumonia)	IV, then oral steroid, IvIg Methotrexate and Hydroxychloroquine
York		Dose 2, May 2021			
Female, 68		(34 weeks)			
Case 2	No	AZD1222/ChAdOx1	Asthenia, impaired swallowing. CK > 7000 U/L and CRP 24 mg/L, no biopsy. widespread muscle oedema in thighs on MRI ANA + Myoblot positive (Mi2)	Gottron’s papules; heliotrope rash; shawl’s sign	As above
York		Dose 2, May 2021			
Female, 68		(25 weeks)			
Case 3	No	AZD1222/ChAdOx1	Amyopathic, sicca symptoms, shortness of breath. CRP 20 mg/L, CK normal. ANA + Myoblot + (SL 75, Ro52)	Heliotrope rash, mechanics hands. Interstitial lung disease. Pericardic effusion	Oral steroids, then IV cyclophosphamide followed by mycophenolate
York		dose 1, May 2021			
Female, 58		(4 weeks)			
Case 4	No	AZD1222/ChAdOx1	Asthenia, general malaise, weight loss. CK 17,000 U/L, CRP normal, myositic pattern on EMG. ANA negative, Myoblot negative	No	Oral steroids and methotrexate
Scarborough		dose 1, March 2021			
Male, 61		(2 weeks)			
Case 5	No	BNT162b2	CK 4793 U/L, CRP 55 mg/L. Anti-Jo1 + (>8.0 U/L) Ro52 1.00 (cut-off = <0.99)	Usual interstitial pneumonia (cataract, iatrogenic)	Oral steroids
Harrogate		Dose 3, September 2021			
Male, 82		(4 weeks)			
Case 6	No	AZD1222/ChAdOx1	Asthenia, myalgia. CK 7598 U/L, CRP normal, myositic pattern on EMG, muscle oedema on MRI, biopsy positive ANA negative Myoblot + (Pl12 and Scl100)	No	Steroids, then azathioprine and one IV drip of immunoglobulins
Bradford		dose 2, February 2021			
Female, 76		(5 weeks)			
Case 7	No	BNT162b2	Asthenia, myalgia. CK 15,212 U/L, CRP 13 mg/L, muscle oedema on MRI, necrosis-related features on biopsy ANA negative, Myoblot negative Anti-HMGCR +	No	Steroids, then two IV drips of immunoglobulins
Bradford		dose 3, November 2021			
Male, 64		(5 weeks)			
Case 8	No	BNT162b2	Asthenia, myalgia. CK 8038 U/L, CRP normal, muscle oedema on MRI ANA negative, Myoblot negative	No	Physiotherapy
Bradford		dose 2, May 2021			
Male, 70		(24 weeks)			
Case 9	Systemic lupus erythematosus (CK normal, no clinical myositis)	AZD1222/ChAdOx1	Asthenia, myalgia. CK 1299 U/L, CRP normal, myositic pattern on EMG, muscle oedema on MRI, biopsy positive ANA + (Ro/RNP/Sm/ribosomal)	No (Immune thrombocytopenic purpura causing stroke)	Oral steroids and methotrexate
Bradford		dose 1, January 2021			
Female, 37		(4 weeks)			
Case 10	No	BNT162b2	Asthenia. CK 3581 U/L, CRP 109 mg/L, myositic pattern on EMG, muscle oedema on MRI, biopsy positive ANA negative, Myoblot negative Anti-HMGCR +	Mild interstitial changes on computed scan, asymptomatic	IV, then oral steroid Methotrexate
Mid-Yorkshire		Dose 3, October 2021			
Female, 71		(5 weeks)			
Case 11	No	AZD1222/ChAdOx1	Asthenia, myalgia. CK 2725 U/L, CRP 53 mg/L, muscle oedema on MRI, biopsy negative ANA + (Jo1)	Pulmonary embolism	Oral steroids and azathioprine (not tolerated), then methotrexate
Hull		dose 1, January 2021			
Female, 78		(2 weeks)			
Case 12	No	AZD1222/ChAdOx1	Asthenia, dysphagia, respiratory arrest. CK 149,430 U/L, CRP 220 mg/L, Necrosis on biopsy ANA + (Ro/La) Myoblot negative Anti-HMGCR negative	Anuria (renal failure haemodialysis-dependant), respiratory arrest (dependent on intensive care support), suspected myocarditis. Multiple supra-infections. Ultimately death	IV steroids and IV drips of immunoglobulins and rituximab
Hull		dose 2, May 2021			
Male, 67		(6 weeks)			
Case 13	No	BNT162b2	Asthenia. CK 3654 U/L, normal CRP, myositic pattern on EMG, muscle oedema on MRI. Biopsy not performed (declined by patient. ANA negative, Myoblot negative Anti-HMGCR +	No	Physiotherapy
Mid-Yorkshire		dose 1, May 2021			
Female, 72		(<1 week)			
Case 14	No	BNT162b2	Asthenia. CK 5602 U/L, normal CRP, myositic pattern on EMG, muscle oedema on MRI. Biopsy slighty suggestive of myositis (drying artifacts). ANA + (Sm/RNP/anti-chromatin+)	Raynaud’s phenomenon	Oral steroids and methotrexate
Leeds		dose 2, August 2021			
Female, 37		(12 weeks)			
Case 15	Giant Cell Arteritis	AZD1222/ChAdOx1	Asthenia, general malaise, weight loss. CK 3070 U/L, CRP 19.5 mg/L, myositic pattern on EMG, muscle oedema on MRI, Necrosis on biopsy. ANA negative, Myoblot + (anti-SRP)	No	Oral steroids and methotrexate
Leeds		dose 2, March 2021			
Female, 83		(6 weeks)			

IMID = Immune-Mediated Inflammatory Disease (e.g.,: rheumatoid arthritis, systemic lupus erythematosus); CRP = C-reactive protein; CK = Creatine-kinase; MRI = Magnetic resonance imaging; ANA = Anti-Nuclear autoantibodies; IV = intra-venous; EMG = electromyography; Anti- HMGCR = anti 3-hydroxy-3-methylglutaryl-coenzyme A reductase antibodies.

**Table 2 vaccines-10-01184-t002:** Myositis Cases Therapy and Responses.

KERRYPNX	Relevant Medications before Myositis Onset	Duration of Symptoms Severity	Amounts of Medications Administered to Treat Myositis	Duration of Myositis treatment	Interval to Recovery (If Applicable)
Case 1	No other vaccinations preceding exposure to AZD1222/ChAdOx1	8 weeks	1 IV pulse of MP (1.5 g) followed by oral prednisolone 35 mg/day; both associated with IVIG (150 g), MTX 25 mg/week and HCQ 400 mg/day	7 months	3 months (muscular strength recovery) Swallow unrecovered
	Not on relevant medications (including statins)				
Case 2	No other vaccinations preceding exposure to AZD1222/ChAdOx1	10 days	1 IV pulse of MP (1.5 g) followed by oral prednisolone 40 mg/day (tapered); both associated with IVIG (90 g), MTX 20 mg/week and HCQ 200 mg/day	4 months	4 months
	Not on relevant medications (including statins)				
Case 3	No other vaccinations preceding exposure to AZD1222/ChAdOx1	7 days	Oral prednisolone 40 mg/day (tapered), associated with CYCLO 6 IV pulses (15 mg/kg) and MMP 2 g/day	3 months	3 months (partial muscular strength recovery, residual fatigue)
	On levothyroxine 100 mcg/day				
	Not on statins				
Case 4	No other vaccinations preceding exposure to AZD1222/ChAdOx1	10 months	Oral prednisolone 30 mg/day (tapered to 0) MTX 20 mg/week	6 months	6 months (partial recovery of muscular strength)
	Not on relevant medications (including statins)				
Case 5	Flu vaccine was received at the same time of exposure to BNT162b2	2 months	Oral prednisolone 40 mg/day (tapered to 0) MMP 2 g/day	2 months	2 months
	Simvastatin (stopped at the time of myositis onset)				
Case 6	No other vaccinations preceding exposure to AZD1222/ChAdOx1	21 days	oral prednisolone 60 mg/day (tapered to 0); associated with both IVIG (90 g) and AZA 2.5 mg/kg	4 months	4 months
	Atorvastatin (stopped at admission)				
Case 7	No other vaccinations preceding exposure to BNT162b2	45 days	oral prednisolone 60 mg/day (tapered to 0); associated with both IVIG (120 g) and MTX 25 mg/week	6.5 months	4 months
	Atorvastatin (stopped at admission)				
Case 8	No other vaccinations preceding exposure to BNT162b2	10 days	Conservative approach and physiotherapy	5 months	3 months
	Not on relevant medications (including statins)				
Case 9	No other vaccinations preceding exposure to AZD1222/ChAdOx1	5 weeks	oral prednisolone 60 mg/day (initially tapered to 0); associated with MTX 25 mg/week (stopped due to ITP), then with RTX (two doses of 1 g two weeks apart)	7 months	2 months (transient recovery, then relapse of myositis; treatment ongoing)
	On azathioprine for SLE				
Case 10	No other vaccinations preceding exposure to BNT162b2	2 weeks	3 IV pulses of MP (1 g) followed by oral prednisolone 60 mg/day (tapered to 0); both associated with MTX 25 mg/week	5 months	1.5 months
	Atorvastatin (stopped at admission)				
Case 11	No other vaccinations preceding exposure to AZD1222/ChAdOx1	6 weeks	oral prednisolone 60 mg/day (tapered to 0); associated with MTX 25 mg/week	5.5 months	3 months
	Not on relevant medications (including statins)				
Case 12	No other vaccinations preceding exposure to AZD1222/ChAdOx1	2 months	3 IV pulses of MP (1 g) followed by IV hydrocortisone 300 mg/day; both associated with IVIG (165 g) and RTX (two doses of 1 g two weeks apart)	2 months	Not applicable (death)
	Atorvastatin (stopped at admission)				
Case 13	No other vaccinations preceding exposure to BNT162b2	2 months	None	Not applicable	2 months
	Atorvastatin (stopped at myositis onset)				
Case 14	No other vaccinations preceding exposure to BNT162b2	6 months	Oral prednisolone 60 mg/day (tapered to 45); Associated with MTX 20 mg/week	2 months	2 months (full recovery not achieved)
	Not on relevant medications (including statins)				
Case 15	No other vaccinations preceding exposure to AZD1222/ChAdOx1	12 months	Oral prednisolone 40 mg/day (tapered to 15); Associated with MTX 20 mg/week	3.5 months	3.5 months (full recovery not achieved)
	Not on relevant medications (including statins)				

IV = intravenous; MP = methylprednisolone; IVIG = intravenous immunoglobulins; MTX = methotrexate; HCQ = hydroxychloroquine; CYCLO = cyclophosphamide; MMP = mophetil mychophenolate; AZA = azathioprine; ITP = Immune Thrombocytopenic Purpura; RTX = rituximab.

## Data Availability

Not applicable.

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
