# Peer review of "A Large Cluster of New Onset Autoimmune Myositis in the Yorkshire Region Following SARS-CoV-2 Vaccination"

_vaccines, 2022, doi:10.3390/vaccines10081184_

Round 1
Reviewer 1 Report
I wonder, why the authors do not exclude patient number 9 too. she already has positive results for anti-Ro/RNP and anti-chromatin autoantibodies prior to exposure to vaccines and myositis development, and had previously been treated as inflammatory arthritis and was treated with azathioprine years before vaccination and could represent an undefined connective tissue disease prior to definitive myositis diagnosis post-vaccination?. Because she is young (37 y) and already SLE, please considered this recent review report for more details (PMID: 35040083) that anti-Ro circulating in 20-40% of autoimmune myositis, you cannot exclude she had pre autoimmune myositis and you do not mention if she had CK before vaccination.
Many published reports already agree with your plausible explanation for muscle autoimmunity, so please find and cite them to support your expectation which needs more intensive molecular immunology analysis.
Author Response
COMMENT - I wonder, why the authors do not exclude patient number 9 too. she already has positive results for anti-Ro/RNP and anti-chromatin autoantibodies prior to exposure to vaccines and myositis development, and had previously been treated as inflammatory arthritis and was treated with azathioprine years before vaccination and could represent an undefined connective tissue disease prior to definitive myositis diagnosis post-vaccination?. Because she is young (37 y) and already SLE, please considered this recent review report for more details (PMID: 35040083) that anti-Ro circulating in 20-40% of autoimmune myositis, you cannot exclude she had pre autoimmune myositis and you do not mention if she had CK before vaccination.
RESPONSE - We suspect that the temporal onset of myositis may be due to the fact that some patients are at imminent risk and that RNA and DNA vectored vaccines may provide the non specific adjuvant stimulus to move from normal muscle function to myositis. Accordingly, we would like to keep the SLE case that had no prior myositis (either clinical or by laboratory findings. We agree with the comments about the possibility of early prior undiagnosed myositis but this could apply to many of the cases.
COMMENT - Many published reports already agree with your plausible explanation for muscle autoimmunity, so please find and cite them to support your expectation which needs more intensive molecular immunology analysis.
RESPONSE - We are grateful for this comment and we have expanded the references accordingly.
Reviewer 2 Report
Potential adverse reactions to SARS vaccines is an important topic. The presentation of potential adverse reaction by the media and by some healthcare workers has inhibited many members of the public from accepting vaccination.
My comments below are to assist you in presenting your conclusions as clearly as possible and minimising the possibility of misunderstanding your research. Your table would benefit by significant editing for clarity.
Abstract
“x/15 had myositis associated autoantibodies.” [please check]
Results
1. Could you please present a table of the illnesses, medications, (and in this case other vaccines) of your patients. E.g., you wrote for one patient “Six of 15 cases were on statins at the time of myositis onset.”
2. Could you please provide numerical results for the lab data. E.g., Case 1. You entered these data: “Deranged CK and CRP, no biopsy. ANA +, Myoblot positive (SAE-1). Please replace “Deranged” with numerical data. I have never seen this word applied to describe lab data.
3. Could you please assemble your data in a table to demonstrate more clearly which findings support and which do not support your diagnoses for each patient.
Two examples:
E.g. you wrote: “Myositis-associated autoantibodies were positive in 10 cases and negative in 5 cases.”
[Please state how you assess how the 5 negative tests affect the diagnosis]
Only one patient (no 15) had a prior history of autoimmune disease. This is strong evidence for your hypotheses.
4. Please provide details for each patient of the duration of severity of symptoms, amounts of medications and duration of treatment and interval to recovery so that the reader can assess the patients’ clinical courses.
5. You provide an estimate of annual autoimmune myositis cases of 1.19 to 19/million. For Yorkshire this would be 6.5 to 107. Please provide data for your case rate for previous years, perhaps using interquartile data.
Author Response
COMMENT - Potential adverse reactions to SARS vaccines is an important topic. The presentation of potential adverse reaction by the media and by some healthcare workers has inhibited many members of the public from accepting vaccination.
RESPONSE - Many thanks. We fully agree with the referee about the presentation of potential adverse effects might influence vaccine uptake. However, the reporting of events following vaccination is one important mean at the disposal of scientific community to raise awareness of potentially uncommon unwanted effects of the intervention. Indeed, the emergence of VITT and the restriction of DNA vaccines in younger subjects comes from the health care workers and the media coverage of such matter, leading to improved vaccine safety.
COMMENT - My comments below are to assist you in presenting your conclusions as clearly as possible and minimising the possibility of misunderstanding your research. Your table would benefit by significant editing for clarity.
REPOSNSE - Thank you. All sections of the manuscript underwent revision to address inaccuracies. We have edited the Table 1 to enhance clarity. Also, we expanded the presented data by providing one additional table (table 2) that enrich clinical information related to vaccination history, therapies for myositis and clinical course (severity and time-to-recovery/extent of recovery).
COMMENT - Abstract - “x/15 had myositis associated autoantibodies.”
RESPONSE – Many thanks. The section you highlighted underwent revision to address inaccuracies.
COMMENT - Results -
- Could you please present a table of the illnesses, medications, (and in this case other vaccines) of your patients. E.g., you wrote for one patient “Six of 15 cases were on statins at the time of myositis onset.”
RESPONSE - We have expanded this data as suggested and produced one additional table.
COMMENT - 2. Could you please provide numerical results for the lab data. E.g., Case 1. You entered these data: “Deranged CK and CRP, no biopsy. ANA +, Myoblot positive (SAE-1). Please replace “Deranged” with numerical data. I have never seen this word applied to describe lab data.
RESPONSE - We have added specific laboratory values for the patients
COMMENT - 3. Could you please assemble your data in a table to demonstrate more clearly which findings support and which do not support your diagnoses for each patient.
Two examples:
E.g. you wrote: “Myositis-associated autoantibodies were positive in 10 cases and negative in 5 cases.” [Please state how you assess how the 5 negative tests affect the diagnosis]
RESPONSE - The myositis associated autoantibodies supported the diagnosis whenever positive. However, the diagnosis of myositis in our series remains clinical based and investigations-aided, according to the judgement of treating physicians.
COMMENT - Only one patient (no 15) had a prior history of autoimmune disease. This is strong evidence for your hypotheses.
RESPONSE - Indeed, we agree. The idea is that autoimmune prone subjects can rapidly evolve from potential subclinical muscle autoimmunity to muscle disease.
Page 1 of 2
COMMENT - 4. Please provide details for each patient of the duration of severity of symptoms, amounts of medications and duration of treatment and interval to recovery so that the reader can assess the patients’ clinical courses.
REPOSNSE - Many thanks, we produced a new table 2 to report on the variables requested by the reviewer.
COMMENT - 5. You provide an estimate of annual autoimmune myositis cases of 1.19 to 19/million. For Yorkshire this would be 6.5 to 107. Please provide data for your case rate for previous years, perhaps using interquartile data.
RESPONSE – Thank you, this is a good point but unfortunately we do not have a joined up Yorkshire myositis database. This work was based on the uncommonly large numbers of cases turning up.
Reviewer 3 Report
Estimated Authors,
Estimated Editors,
in the present brief article from De Marco et al, a series of 15 cases of myositis from Yorkshire have been reported and discussed, with accurate comparisons with National and international estimates. Briefly, these increased reporting is scarcely consistent with pre-existent data on myositis epidemiology, and the sequential relationship between vaccine and myositis may be indicative of a potential long-term cause.
In fact, as reported: "New onset disease occurred after first vaccination (5 cases), second vaccination (7 cases) or after third dose (3 cases) which was often a different vaccine. Of the cases, 6 had systemic complications including skin (3 cases), lung (3 cases), heart (2 cases) and x/15 had myositis associated autoantibodies".
Such results, deserves some attentions from the public health authorities, and represent a sort of re-enforcement for promoting a quick acceptance of this study.
Author Response
COMMENT - In the present brief article from De Marco et al, a series of 15 cases of myositis from Yorkshire have been reported and discussed, with accurate comparisons with National and international estimates. Briefly, these increased reporting is scarcely consistent with pre-existent data on myositis epidemiology, and the sequential relationship between vaccine and myositis may be indicative of a potential long-term cause.
RESPONSE - We thank the referee for this comment which captures the essence of our work.
COMMENT - In fact, as reported: "New onset disease occurred after first vaccination (5 cases), second vaccination (7 cases) or after third dose (3 cases) which was often a different vaccine. Of the cases, 6 had systemic complications including skin (3 cases), lung (3 cases), heart (2 cases) and x/15 had myositis associated autoantibodies".
Such results, deserves some attentions from the public health authorities, and represent a sort of re- enforcement for promoting a quick acceptance of this study.
RESPONSE - We fully agree with these sentiments and we have down our best to turn the paper around quickly to facilate fast publication. We have also provided a mechanistic model in a new Figure 4 to offer a model for the findings. This can now be tested in the real world setting in databases and in animal models.
Round 2
Reviewer 2 Report
Thanks to the authors for a comprehensive and careful update. This will be of great help to subsequent authors and eventually systematic reviewers summarising a series of such case reports.
Author Response
Many thanks for your reply.
We have also re-edited the manuscript according to requests from the scientific editor and we are grateful to them and you for all the precious suggestions provided.
With regards,
G. De Marco and D. McGonagle